# Analysis of Promoter Methylation of the Bovine *FOXO1* Gene and Its Effect on Proliferation and Differentiation of Myoblasts

**DOI:** 10.3390/ani13020319

**Published:** 2023-01-16

**Authors:** Pengfei Shi, Yong Ruan, Wenjiao Liu, Jinkui Sun, Jiali Xu, Houqiang Xu

**Affiliations:** 1Key Laboratory of Animal Genetics, Breeding and Reproduction in the Plateau Mountainous Region, Ministry of Education, Guizhou University, Guiyang 550025, China; 2College of Animal Science, Guizhou University, Guiyang 550025, China

**Keywords:** *FOXO1*, methylation, myoblasts, proliferation, differentiation

## Abstract

**Simple Summary:**

The purpose of this study was to initially determine the role of methylation of the promoter region of Forkhead box O 1 (*FOXO1*) in regulating its transcriptional level and to further investigate the effect of *FOXO1* on the proliferation and differentiation of bovine myogenic cells. In this study, we used bisulfite sequencing polymerase chain reaction, real-time quantitative PCR, western blot, cell counting kit-8 (CCK-8), and flow cytometry and found that the mRNA expression of the *FOXO1* was low when the methylation of *FOXO1* promoter region was high, and silencing the expression of the *FOXO1* gene could promote the proliferation and differentiation of myoblasts.

**Abstract:**

This study aimed to explore the regulatory role of *FOXO1* promoter methylation on its transcriptional level and unravel the effect of *FOXO1* on the proliferation and differentiation of bovine myoblasts. Bisulfite sequencing polymerase chain reaction (BSP) and real-time quantitative PCR were performed to determine the methylation status and transcript levels of the *FOXO1* promoter region at different growth stages. BSP results showed that the methylation level in the calf bovine (CB) group was significantly higher than that in the adult bovine (AB) group (*p* < 0.05). On the other hand, qRT-PCR results indicated that the mRNA expression level in the AB group was significantly higher than that in the CB group (*p* < 0.05), suggesting a significant decrease in gene expression at high levels of DNA methylation. CCK-8 and flow cytometry were applied to determine the effect of silencing the *FOXO1* gene on the proliferation of bovine myoblasts. Furthermore, qRT-PCR and Western blot were conducted to analyze the expression of genes associated with the proliferation and differentiation of bovine myoblasts. Results from CCK-8 revealed that the short hairpin *FOXO1* (sh*FOXO1*) group significantly promoted the proliferation of myoblasts compared to the short-hairpin negative control (shNC) group (*p* < 0.05). Flow cytometry results showed a significant decrease in the number of the G1 phase cells (*p* < 0.05) and a significant increase in the number of the S phase cells (*p* < 0.05) in the sh*FOXO1* group compared to the shNC group. In addition, the expression of key genes for myoblast proliferation (*CDK2*, *PCNA*, and *CCND1*) and differentiation (*MYOG*, *MYOD*, and *MYHC*) was significantly increased at both mRNA and protein levels (*p* < 0.05). In summary, this study has demonstrated that *FOXO1* transcription is regulated by methylation in the promoter region and that silencing *FOXO1* promotes the proliferation and differentiation of bovine myoblasts. Overall, our findings lay the foundation for further studies on the regulatory role of epigenetics in the development of bovine myoblasts.

## 1. Introduction

Muscle proliferation and differentiation, a highly coordinated and complex biological process, have been extensively studied at the genetic level. Studies have demonstrated that several vital genes are involved in its regulation, including the cell cyclin gene family (*CCN*), the cyclin-dependent kinase gene family (*CDK*), the proliferating cell nuclear antigen (*PCNA*), and myogenic regulatory factors (*MRF*) [1,2,3]. However, there is a need for an in-depth understanding of the regulatory mechanisms of muscle proliferation and differentiation, with the overarching goal of improving the economic benefits of meat.

The mammalian Forkhead Box O (FOXO) transcription factor family comprises five members: *FOXO1*, *FOXO3*, *FOXO3b*, *FOXO4*, and *FOXO6* [4]. *FOXO1*, the first member to be identified, acts as an essential mammalian transcription factor that plays a regulatory role in various cells [5]. Most early studies on *FOXO1* were associated with cancer therapy since it has been considered a tumor suppressor mainly due to its potential to promote apoptosis [6]. Studies have found that *FOXO1* deficiency causes lethal defects in mouse embryonic angiogenesis [7,8], suggesting that it modulates cell growth and development. In addition, numerous studies have reported that *FOXO1* regulates myoblasts, ovarian granulosa cells, and adipocytes, and its activation in cells can lead to cell cycle arrest and subsequent apoptosis [9,10]. Qi et al. [11] regulated *FOXO1* gene expression in sheep ovarian granulosa cells and found that the gene inhibited the proliferation of sheep GCs and affected the mRNA expression abundance of genes involved in the regulation of apoptosis, cell cycle, and steroid synthesis. It has also been reported that abundant expression of *FOXO1* in adipocytes regulates the expression of genes associated with adipocyte differentiation, anti-oxidative stress, and lipid metabolism and modulates the adipocyte size to adapt to calorie intake, thereby regulating energy balance [12,13]. A previous study revealed that overexpression of *FOXO1* significantly inhibits the differentiation process of C2C12 myoblasts [14]. Moreover, inhibition of myogenic differentiation caused by *FOXO1* was significantly restored by insulin treatment since it is one of the most important transcription factors in the insulin/insulin-like growth factor 1 (*IGF-1*) signaling pathway [15]. Although *FOXO1* plays an inhibitory role in the early differentiation of myoblasts, it can stimulate myotube fusion in primary mouse myoblasts at the end of differentiation [9,16]. Studies have also shown that the expression of *FOXO1* increases gradually during the differentiation of primary bovine myoblasts, and its polymorphism is associated with the productive performance of yellow cattle [17,18].

Mammalian DNA methylation is an important epigenetic modification that is dynamically regulated during development [19]. Its ability to modulate cell proliferation and differentiation by regulating the timing and level of gene expression, and excessive aberrant methylation can lead to cellular carcinogenesis [20,21]. Evidence suggests that DNA methylation in the promoter region negatively regulates mRNA expression [22]. The main aim of this study was to determine the role of methylation in the promoter region of *FOXO1* in regulating its transcriptional level and to further explore the effect of *FOXO1* on the proliferation and differentiation of bovine myoblasts. Collectively, our findings provide fundamental evidence to unravel the regulatory mechanisms of *FOXO1* at the transcriptional level and its effects on the function of myoblasts.

## 2. Materials and Methods

### 2.1. Tissue Sample Collection

We randomly selected 3-day-old (calf bovine, CB) and 24-month-old (adult bovine, AB) healthy Guanling cattle under the same feeding conditions (Guanling Cattle Industrial Park, Guanling County, Anshun City, Guizhou Province, China). After slaughter by Guanling Cattle Industrial Park staff, the longissimus dorsi muscle was harvested, and the separated tissue samples were stored in liquid nitrogen for backup. The use of experimental animals in this study was approved by the Laboratory Animal Ethics of Guizhou University (No. EAE-GZU-2021-E019, Guiyang, China; 1 November 2021).

### 2.2. Bisulfite Sequencing Polymerase Chain Reaction

Genomic DNA was extracted from the tissue samples using the OMEGA (USA) DNA Extraction Kit according to the manufacturer’s instructions. The extracted DNA was treated with sodium bisulfite using the EZ DNA Methylation-Gold Kit (Zymo Research, Orange County, CA, USA), following the manufacturer’s instructions. Next, each unmethylated cytosine in the chemically transformed DNA was converted to uracil, whereas each methylated cytosine was left unchanged. The methylation status of the FOXO1 promoter region was subsequently analyzed by the bisulfite sequencing polymerase chain reaction (BSP). CpG islands in the *FOXO1* promoter region (Gene ID: 506618) were predicted by the MethPrimer (http://www.urogene.org/cgi-bin/methprimer/methprimer.cgi, accessed on 23 June 2021), followed by the design of BSP primers using Primer Premier 5.0 (Table 1). The treated DNA was amplified using the Hieff^®^ Hotstart PCR Genotyping Master Mix (YEASEN, Shanghai, China) in a Bio-Rad thermal cycler. The following thermocycling conditions were applied: pre-denaturation for 5 min at 95 °C for one cycle; followed by denaturation for 30 s at 95 °C, annealing for 30 s at 51 °C, and extension for 40 s at 72 °C for 35 cycles; and final extension for 10 min at 72 °C. The BSP products were purified using an agarose gel DNA extraction and purification kit (OMEGA, Norcross, GA, USA) according to the manufacturer’s instructions and then ligated with the pMD19-T vector (TaKaRa, Dalian, China). Finally, 10 positive clones from each sample were randomly selected for sequencing (Qingke, Beijing, China).

### 2.3. Real-Time Quantitative PCR

Total RNA was extracted from the tissue samples and cells using the TRIzol reagent (Solarbio, Beijing, China). Next, first-strand cDNA synthesis was performed using the StarScrip II First Strand cDNA Kit (Genstar, Beijing, China) according to the manufacturer’s instructions, and the reaction products were stored at −20 °C. The qRT-PCR was then performed to analyze the relative expression of FOXO1, value-added key genes (CDK2, PCNA, and CCND1), and differentiation key genes (MYOD, MYOG, and MYHC) at the mRNA level. The details of the qRT-PCR primers are shown in Table 1. The qRT-PCR was performed with 2 × PowerUp™ SYBR™ Green Master Mix in a Bio-Rad CFX96™ (Thermo Fisher Scientific, Waltham, MA, USA) real-time detection system according to the manufacturer’s instructions. The following qRT-PCR procedure was applied: pre-denaturation for 2 min at 95 °C for one cycle, followed by denaturation for 15 s at 95 °C, annealing for 15 s at 60 °C, and extension for 30 s at 72 °C for 40 cycles. The final melting curve was created by heating in 0.5 °C steps from 60 °C to 95 °C, and the fluorescence acquisition time was 5 s. Notably, each sample was replicated three times. Relative mRNA expression was normalized to β-actin (ACTB) mRNA as an internal reference and calculated using the 2^−∆∆CT^ quantification method [23].

### 2.4. Plasmid Construction

Multiple pairs of *FOXO1* interfering sequences were designed following the shRNA design principles and sent to GEMA Ltd. (Shanghai, China) for synthesis and tested for interference efficiency by qRT-PCR and Western blot assay. The detailed shRNA target sequences are shown in Table 2.

### 2.5. Cell Culture and Transfection

Bovine myoblasts were isolated from calf longissimus dorsi muscle by enzymatic digestion, and cell purity was analyzed by indirect immunofluorescence [24,25]. The sorted myoblasts were inoculated in six-well plates using DMEM-F/12 (Gibco, San Diego, CA, USA), which contains 15% fetal bovine serum and 1% penicillin-streptomycin. When myoblasts reached 80% confluence, they were transfected with sh*FOXO1* and shNC using Lipofectamine 3000 (Thermo Fisher Scientific, Waltham, MA, USA) transfection reagent according to the manufacturer’s instructions. After 48 h of transient transfection, cells were harvested for subsequent analysis, including qRT-PCR, Western blot analysis to determine cell proliferation, and flow cytometry for cell cycle analysis. With regard to cell differentiation studies, the medium was changed to differentiation medium (DMEM-F/12 + 2% horse serum) 6 h after completing transient transfection to induce cell differentiation. Cells were incubated for 48 h and then harvested for qRT-PCR and Western blot analyses.

### 2.6. Western Blot

Total protein was extracted by treating collected cells with RIPA lysate (Solarbio, Beijing, China) containing protease inhibitors, followed by measuring the protein concentration using a BCA protein concentration assay kit (Solarbio, Beijing, China). Equal protein samples were resolved using 10% SDS-PAGE gel electrophoresis and transferred to PVDF membranes. Next, membranes were blocked with 5% skimmed milk for 2 h and incubated with primary antibodies overnight at 4 °C. The following primary antibodies were used: FOXO1 (Rabbit anti-FOXO1, ABclonal, Wuhan, China), PCNA (Rabbit anti-PCNA, Proteintech, Wuhan, China), CCND1 (Rabbit anti-CCND1, Proteintech, Wuhan, China), CDK2 (Rabbit anti-CDK2, Bioss, Beijing, China), MYOD (Rabbit anti-MYOD, Bioss, Beijing, China), MYOG (Rabbit anti-MYOG, ABclonal, Wuhan, China), MYHC (Rabbit anti-MYHC, ABclonal, Wuhan, China), and ACTB (Rabbit anti-ACTB, Proteintech, Beijing, China). On the next day, membranes were washed three times with Tris-buffered saline with Tween-20 (TBST) buffer and incubated with HRP-labelled secondary antibodies for 2 h at 37 °C. Membranes were washed three times with TBST and visualized using ECL (NeoSami, Suzhou, China) ultrasensitive luminescence. Finally, images were captured using the ChemiDocXRS system, and the bands were analyzed in greyscale using Image J software.

### 2.7. Cell Counting Kit-8 (CCK-8) Assay to Determine Cell Proliferation

The CCK-8 assay was performed to explore the effect of *FOXO1* gene silencing on the proliferative activity of myoblasts. Briefly, myoblasts were inoculated on 96-well plates, transfected with the negative control and test groups, and then incubated at 37 °C and 5% CO_2_ for 12 h, 24 h, 48 h, and 72 h according to the instructions of the CCK-8 reagent (APExBIO, Houston, TX, USA). Finally, the absorbance (OD) was measured at 450 nm using an enzyme marker (Thermo Fisher Scientific, Waltham, MA, USA).

### 2.8. Flow Cytometry to Determine the Cell Cycle

Briefly, myoblasts were fixed overnight at 4 °C using pre-cooled 70% ethanol (7:3 ratio of ethanol to PBS) and then treated with a cell cycle assay kit (Servicebio, Wuhan, China) according to the manufacturer’s instructions. The DNA content of cells under different treatment conditions was then measured using a flow cytometer (CytoFLEX, Beckman, Brea, CA, USA). Notably, each treatment was replicated three times.

### 2.9. Statistical Analysis

All statistical analyses were performed using SPSS 18.0 software (IBM SPSS Statistics 18, Inc., Chicago, IL, USA), and all data are presented as means ± SD of three biological replicates and three technical replicates to ensure the accuracy of the experimental data. A one-way analysis of variance (ANOVA) was used to compare the differences between the various groups. Statistical significance was considered at *p* < 0.05.

## 3. Results

### 3.1. Effect of FOXO1 Promoter Methylation on Transcript Levels

Analysis of the *FOXO1* promoter region revealed the presence of a total of two CpG islands, with CpG1 (−653~−1077) and CpG2 (−366~−676) containing 43 and 29 methylation sites, respectively (Figure 1A,B). The BSP results showed that the average methylation rates were 0.972%, 1.25%, and 1.25% in the CB group and 0.556%, 0.417%, and 0.556% in the AB group, indicating that the methylation rate of the *FOXO1* promoter region was significantly higher in the CB group than in the AB group (*p* < 0.05) (Figure 1C). To confirm whether the methylation status of the *FOXO1* promoter region affected its expression level in muscle tissue, we examined the expression levels of *FOXO1* mRNA in both groups. The results demonstrated that *FOXO1* was expressed in both groups, but the relative expression of FOXO1 was significantly higher in the AB group compared to the CB group (*p* < 0.05) (Figure 1D), suggesting that high levels of DNA methylation significantly reduced gene expression. The results of the software analysis of the *FOXO1* promoter region and details of the methylation rate of each CpG site are shown in Appendix A.

### 3.2. Cell Purity Assay

The purity of myoblasts was analyzed by indirect immunofluorescence, which showed that myoblasts bound to anti-Desmin fluoresced red, while unbound myoblasts did not. Myoblasts bound to anti-Desmin were observed under different fields of view, and by manual counting, the purity of myoblasts was calculated to be greater than 90% (Figure 2) and thus could be used for subsequent experiments.

### 3.3. Assay of shRNA Interference Efficiency

Firstly, qRT-PCR was applied to the four FOXO1 shRNAs to analyze and screen shRNA1 for the best interference efficiency (Figure 3A). Then it was verified again by Western blot that shRNA1 also has an interference effect at the protein level (Figure 3B), hereafter referred to as sh*FOXO1*.

### 3.4. Effect of Silencing FOXO1 Gene Expression on Myoblast Proliferation

Results revealed that the expression of key genes *(PCNA*, *CDK2*, and *CCND1*) was significantly increased (*p* < 0.05) at both the mRNA and protein levels after silencing the *FOXO1* gene (Figure 4A,B). The CCK-8 assay results showed that the sh*FOXO1* group significantly promoted the value-addition of myoblasts compared to the shNC group (Figure 4C). In addition, flow cytometry results indicated that the number of cells in the G1 phase was significantly reduced (*p* < 0.05), and the number of cells in the S phase was significantly increased (*p* < 0.05) in the sh*FOXO1* group compared to the shNC group (Figure 4D). Altogether, these results suggest that sh*FOXO1* promoted the proliferation of myoblasts.

### 3.5. Effect of Silencing FOXO1 Gene Expression on Myoblast Differentiation

To further explore the effects of *FOXO1* on bovine myoblast differentiation, qRT-PCR and Western blot analyses were applied to determine the relative expression of *MYOD*, *MYOG*, and *MYHC* at both mRNA and protein levels after silencing the *FOXO1* gene. Results demonstrated that the expressions of *MYOD*, *MYOG*, and *MYHC* were significantly elevated at both levels (*p* < 0.05) after silencing the *FOXO1* gene (Figure 5A,B), indicating that sh*FOXO1* promotes differentiation of myoblasts. Details of the original western blot figures are shown in Appendix A.

## 4. Discussion

Muscle development is an essential factor influencing animal growth and a crucial indicator of meat quality. Skeletal muscle development involves a process where myoblasts, differentiated from myosomes, progress toward a myogenic pathway through cell proliferation, terminal differentiation, and fusion into multinucleated muscle fibers [26]. Several studies have revealed that *MYOD* and *MYOG*, muscle-specific transcription factors, are key genes that regulate myoblasts’ fusion [27] and skeletal myogenesis [28,29]. It is worth noting that the skeletal muscle is a dynamic tissue with high contraction and good plasticity. Its composition of *MYHC* isoforms and metabolic activity determine the composition of different muscle fiber isoforms [16]. Muscle development is regulated by many genes whose expression is controlled at either the transcriptional or translational level, thereby affecting the mRNA and protein expression levels. The modification status of DNA methylation, one of the important epigenetic modifications, is dynamic during individual development and participates in the regulation of gene transcription, thereby affecting the expression of genes at the mRNA level [30]. In recent years, epigenetics has become a hotspot in livestock genetic breeding research. It is particularly important to explore the molecular genetic regulatory networks of genes involved in the regulation of muscle growth and development from an epigenetic perspective. In this study, BSP and qRT-PCR were applied to determine the effect of methylation of the *FOXO1* promoter region on transcript levels. BSP results revealed differential methylation of the *FOXO1* promoter region in muscle tissues from the CB and AB groups, whereas qRT-PCR results showed differences in the mRNA expression levels of *FOXO1* at the two growth stages. These results suggest a tendency to significantly reduce gene expression at high DNA methylation levels, which lays the foundation for exploring the function of the differentially methylated region of the *FOXO1* gene.

To further explore the potential function of the *FOXO1* gene, we investigated the effect of *FOXO1* on the proliferation and differentiation of bovine myoblasts through a gene interference strategy. A previous study reported that using cardiotoxin on transgenic mice overexpressing *FOXO1* in the skeletal muscle caused muscle damage and that *FOXO1* reduced the proliferative capacity of myoblasts and disrupted regeneration of the skeletal muscle [31]. In the present study, we found that interference with the *FOXO1* gene caused a significant increase in the expression of the cell proliferation marker gene (*PCNA*), an essential gene for cell proliferation, and a significant increase in the expression abundance of the cell cycle-related proteins CDK2 and CCND1. In contrast, Qi et al. found no significant difference in the relative abundance of *CCND1* and *CCND2* gene transcription products after overexpressing the *FOXO1* gene in sheep granulosa cells [11]. However, *FOXO1* overexpression increased the relative abundance of P21 and P27 proteins, which are important cell cycle protein-dependent kinase inhibitors that can cause cell cycle arrest and proliferation through specific molecular mechanisms [32,33,34]. In addition, another study that explored curcumin-induced *FOXO1* inhibition of lung cancer progression and metastasis found that activation of *FOXO1* inhibited the spread of lung cancer cells by downregulating *CCND1* gene expression [35]. These studies suggest that the *FOXO1* gene regulates cellular proliferation but may affect cellular proliferation through different signaling pathways in different cell types. In this study, CCK-8 and flow cytometry assays were also applied to evaluate the effect of silencing the *FOXO1* gene on myoblasts, with the obtained results validating that the *FOXO1* gene negatively regulates the proliferation of bovine myoblasts.

Previous studies have reported that *FOXO1* inhibits early differentiation of myoblasts and regulates the type of skeletal muscle fibers, with transgenic mice exhibiting significantly reduced skeletal muscle mass, impaired muscle function, and reduced abundance of slow fiber-related gene expression [36,37]. The *FOXO1* protein mainly exerts its function in the nucleus. One study found that when myoblasts undergo multinuclear fusion to form myotubes, the process is associated with the phosphorylation state of *FOXO1* protein, which is translocated to the cytoplasm and loses its regulatory role, suggesting that myoblasts require inhibition of *FOXO1* protein activity during early differentiation [38]. Wu et al. reported that overexpression of the *FOXO1* gene significantly inhibited the differentiation of C2C12 myoblasts [15]. Herein, we found that interference with the *FOXO1* gene significantly increased the expression levels of the *MYOD*, *MYOG*, and *MYHC* genes and promoted the differentiation of myoblasts, which is consistent with Wu et al. Liu et al. found that skeletal muscle weight gain and *MYOD* expression abundance were significantly increased in mice after reducing *FOXO1* expression in C2C12 myoblasts by RNA oligonucleotides [39].

However, one study reported that *FOXO1* is involved in the late differentiation of myoblasts, and its transcript-level expression is gradually upregulated during the differentiation of bovine myoblasts, with a slight but not significant decrease in expression when myotubes are fully formed, suggesting a tendency for *FOXO1* to induce differentiation of bovine skeletal muscle [17]. In addition, Bois et al. demonstrated that *FOXO1* plays a role in cell cycle and apoptosis and controls the fusion rate of myotube formation during mouse myoblast differentiation [40]. This also suggests that *FOXO1* may play a bidirectional regulatory role in the differentiation of myoblasts.

## 5. Conclusions

This study provides preliminary evidence that high methylation levels in the *FOXO1* promoter region are associated with low mRNA expression. However, more data is needed to reveal this trend’s exact impact. Silencing *FOXO1* gene expression promoted the expression of the proliferation key genes (*CDK2*, *PCNA*, and *CCND1*) and the differentiation key genes (*MYOG*, *MYOD*, and *MYHC*) at both mRNA and protein levels. Moreover, CCK-8 and flow cytometry assays further validated that silencing *FOXO1* promotes the proliferation of myoblasts. In summary, this study has revealed that the level of methylation in the promoter region of *FOXO1* leads to changes in its gene expression, which in turn affect the proliferation and differentiation of bovine myoblasts. Overall, our findings provide a theoretical basis for future studies on the regulation of epigenetics in the development of bovine myoblasts.

## Figures and Tables

**Figure 1 animals-13-00319-f001:**
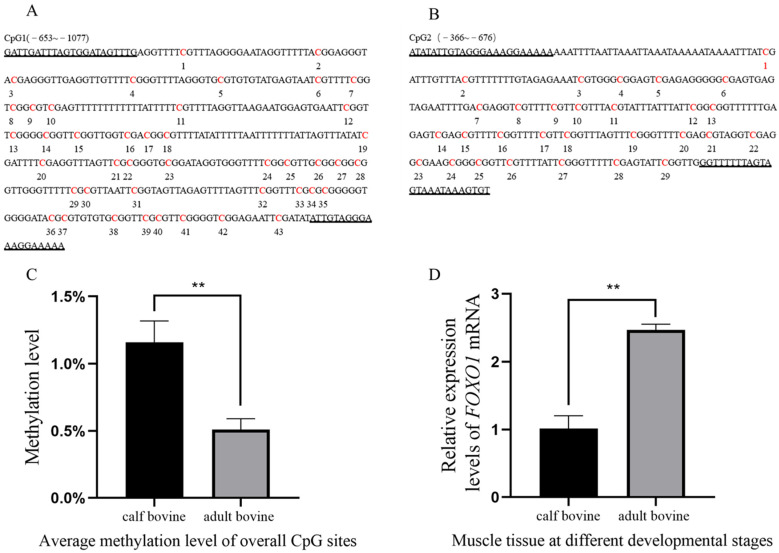
Study on the DNA methylation level of the *FOXO1* gene promoter. (**A**,**B**) CpG islands in the *FOXO1* promoter region (CpG1: −653 to −1077; CpG2: −366 to −676). CpG sites and primer sequences on CpG islands (marked by underlining), where CpG1 and CpG2 contain 43 and 29 CpG sites, respectively. (**C**) Histogram of total methylation levels. Total methylation levels in CB and AB groups were significantly different (*p* < 0.01). (**D**) The mRNA expression levels of *FOXO1* in muscle tissue at different developmental stages were significantly different (*p* < 0.01). Two asterisks (**) indicate significant differences (*p* < 0.01).

**Figure 2 animals-13-00319-f002:**
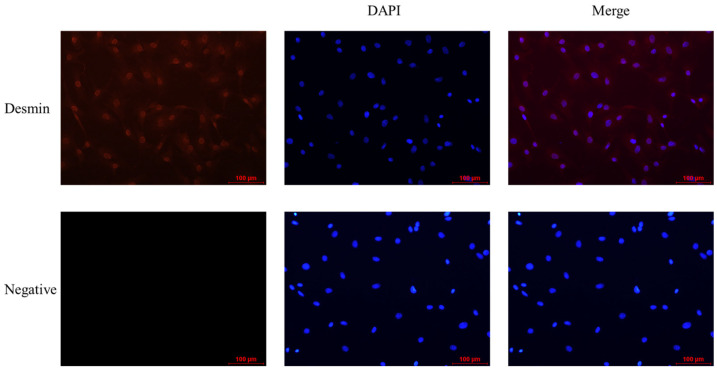
The results of indirect immunofluorescence analysis of myoblasts’ purity were applied with anti-Desmin (red, 1:200, Bioss, Beijing, China), and the nuclei were counterstained with DAPI (blue) (scale bar, 100 μm).

**Figure 3 animals-13-00319-f003:**
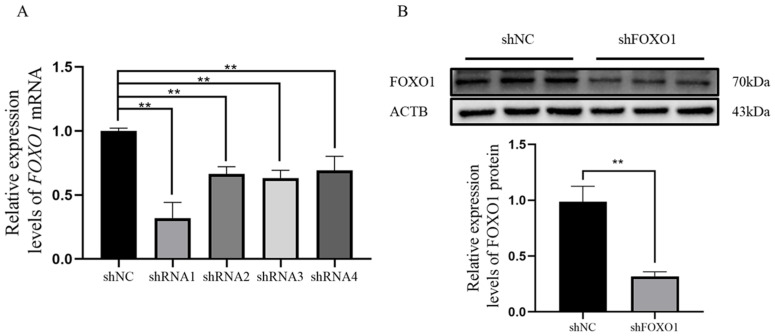
shRNA interference efficiency assay. (**A**,**B**) The qRT-PCR and Western blot assay for interference efficiency of *FOXO1* shRNA. Two asterisks (**) indicate significant differences (*p* < 0.01).

**Figure 4 animals-13-00319-f004:**
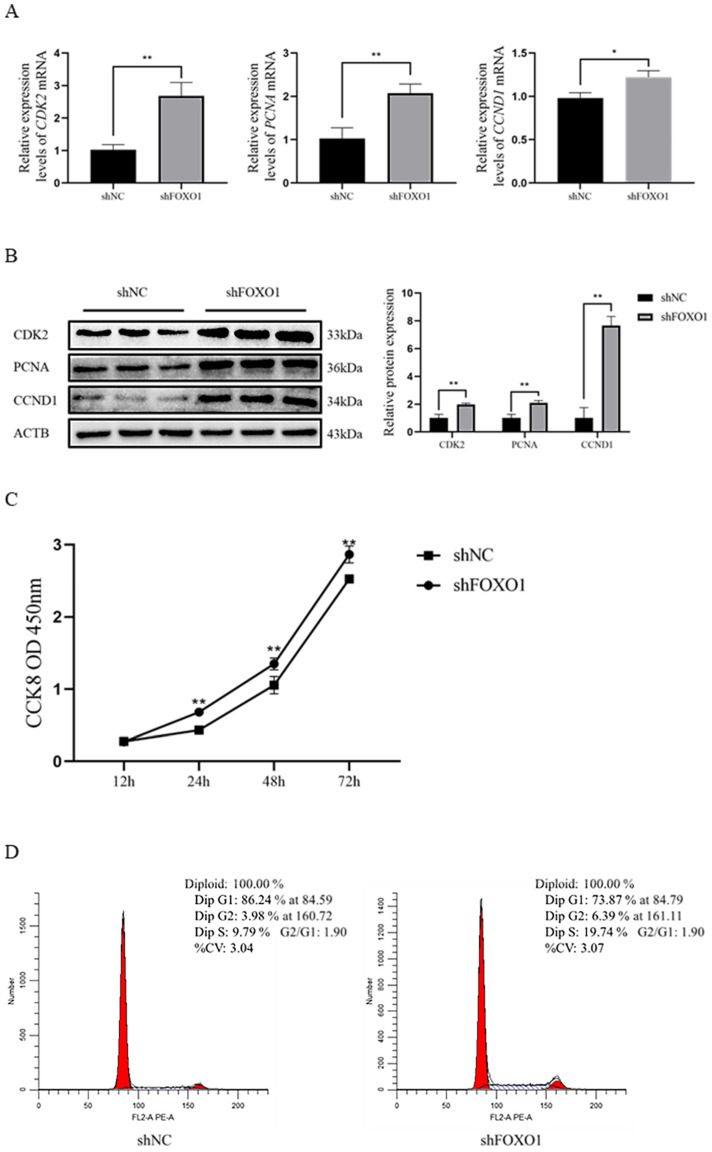
The effect of sh*FOXO1* on the proliferation of myoblasts. (**A**) qRT-PCR to determine the expression of *CDK2*, *PCNA*, and *CCND1* mRNA in the sh*FOXO1* and shNC groups. (**B**) Western blot to detect the expression of CDK2, PCNA, and CCND1 protein in the sh*FOXO1* and shNC groups (greyscale analysis). (**C**) CCK-8 assay was used to detect the proliferation of transfected cells. (**D**) Flow cytometry was used to determine the DNA content of myoblasts 48 h after transfection. An asterisk (*) indicates significant differences (*p* < 0.05), and two asterisks (**) indicate significant differences (*p* < 0.01).

**Figure 5 animals-13-00319-f005:**
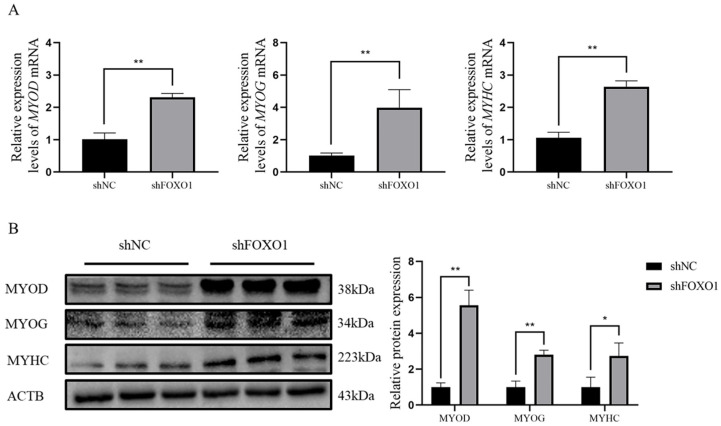
Effect of sh*FOXO1* on myoblast differentiation. (**A**) qRT-PCR to determine the expression of MYOG, MYOD, and MYHC mRNA in sh*FOXO1* and shNC groups. (**B**) Western blot to detect the expression of MYOG, MYOD, and MYHC proteins in sh*FOXO1* and shNC groups (greyscale analysis). An asterisk (*) indicates significant differences (*p* < 0.05), and two asterisks (**) indicate significant differences (*p* < 0.01).

**Table 1 animals-13-00319-t001:** Primers used in this study.

Primer	Primer Sequence (5′-3′)	Accession Numbers	Amplified DNA Fragment (bp)
Primers used in real-time PCR			
*FOXO1*	F: GCAGATTTACGAGTGGATGGTC R: GCAGGGACAGATTATGACGAA	XM_025000053.1	107
*PCNA*	F: ACATCAGCTCAAGTGGCGTGAAC R: GCAGCGGTAAGTGTCGAAGCC	NM_001034494.1	101
*CCND1*	F: CTGGTCCTGGTGAACAAACT R: ACAGAGGGCAACGAAGGT	NM_001046273.2	144
*CDK2*	F: CAAGTTGACGGGAGAAGTGGT R: CTTTATGAGCGGAAGAGGAAT	NM_001014934.1	247
*MYOD*	F: GGCCGCTGTTTACTGTGGG R: CAGCCGCTGGTTTGGGTT	NM_001040478.2	162
*MYOG*	F: TGGGCGTGTAAGGTGTGTAA R: TGCAGGCGCTCTATGTACTG	NM_001111325.1	197
*MYHC*	F: GCCCACTTCTCCCTCATTCACT R: ACCCTTCTTCTTGCCACCTTTC	NM_174117.1	201
*β-actin*	F: ATGATATTGCTGCGCTCGTGG R: TACGAGTCCTTCTGGCCCAT	NM_173979.3	151
Primers used in methylation analysis			
*MF1-1*	F: GATTGATTTAGTGGATAGTTTG R: TTTTTCCTTTCCCTACAAT	Gene ID:506618	424
*MF1-2*	F: ATATATTGTAGGGAAAGGAAAA R: ACACTTTATTTACTACTAAAAAACC		310

Note: F stands for upstream primer, and R stands for downstream primer.

**Table 2 animals-13-00319-t002:** The specific shRNA sequences.

shRNA Name	Sequences
*shRNA1*	5′-3′ CAGTCTGTCCGAGATCAGTAA
*shRNA2*	5′-3′ AGCGGGCTGGAAGAATTCAAT
*shRNA3*	5′-3′ CAGGACAACAAATCGAGTTAT
*shRNA4*	5′-3′ CTGTGACATGGAGTCCATCAT
*shNC*	5′-3′ GTTCTCCGAACGTGTCACGT

## Data Availability

The data presented in this study are available on request from the corresponding author.

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
