# Peer review of "Analysis of Promoter Methylation of the Bovine FOXO1 Gene and Its Effect on Proliferation and Differentiation of Myoblasts"

_animals, 2023, doi:10.3390/ani13020319_

Round 1

Reviewer 1 Report

Aims to unravel the effect of FOXO1 on the proliferation and differentiation of bovine myoblasts, the authors used bisulfite sequencing polymerase chain reaction, Real-Time Quantitative PCR, Western Blot, CCK8, and flow cytometry to explore the regulatory role of FOXO1 promoter methylation on its transcriptional level. The results showed that mRNA expression of FOXO1 was low when the methylation of FOXO1 promoter region was high, and silencing the expression of FOXO1 gene could promote the proliferation and differentiation of myoblasts. Below please find my comments.

1. The Abbreviations that appear for the first time should be written with the full name, such as FOXO1 at Line 14, CCK8 at Line 17, shFOXO1 at Line 31, and so on. The gene name should be formed in italics.

2.  Are the experimental animals (calf bovine and adult bovine) from the same breeds and sex?

3.  Line 136, the period should be removed from “synthesis. and tested”.

4. There are four shRNAs shown in Figure 3A, but only three bands in Figure 3B. Are those three shRNAs randomly selected?

Author Response

Point 1. The Abbreviations that appear for the first time should be written with the full name, such as FOXO1 at Line 14, CCK8 at Line 17, shFOXO1 at Line 31, and so on. The gene name should be formed in italics.

Response 1: Thank you for your suggestion, we checked the full text and completed the revisions.

Point 2. Are the experimental animals (calf bovine and adult bovine) from the same breeds and sex?

Response 2: Yes, the experimental animals (calves and adult cattle) are from the same breeds and sex. They are both male Guanling cattle.

Point 3. Line 136, the period should be removed from “synthesis. and tested”.

Response 3: Thank you for your suggestion, we have revised it in the manuscript.

Point 4. There are four shRNAs shown in Figure 3A, but only three bands in Figure 3B. Are those three shRNAs randomly selected?

Response 4: Figure 3B shows that we first divided the four FOXO1 shRNAs by qRT-PCR to screen the best interference efficiency of shRNA1. Then the interference effect of shRNA1 at the protein level was verified again by Western blot. Also, we have modified this expression in the Results 3.3 section.

Reviewer 2 Report

This works shows that high methylation levels in the FOXO1 promoter region are associated with low mRNA expression, silencing FOXO1, mRNA and protein levels of proliferation or differentiation related genes were increased, which were further  validated by CCK8 and flow cytometry assay in subsequent myocell proliferation experiments, this is an interesting study, the experimental design and results are complete, however I have few questions:

-In your manuscript states that the level of methylation in the promoter region of FOXO1 affects the proliferation and differentiation of bovine myoblastsbut does it directly regulate proliferation and differentiation or indirectly (or in collaboration) with other genes?

- I understand that the methylation levels are different in AB and CB, but it would be nice if proliferation and differentiation levels could be measured after methylation or demethylation in cattle at the same time.

Author Response

Point 1. In your manuscript states that the level of methylation in the promoter region of FOXO1 affects the proliferation and differentiation of bovine myoblasts, but does it directly regulate proliferation and differentiation or indirectly (or in collaboration) with other genes?

Response 1: Thank you for your suggestion. The methylation level of the FOXO1 promoter region affects the proliferation and differentiation of bovine myoblasts indirectly. Our study revealed that the methylation level of the FOXO1 promoter region leads to changes in the expression of its gene at the transcriptional level, which in turn affects the proliferation and differentiation of bovine myoblasts. Also, we have modified the corresponding expression in the Conclusion section.

Point 2. I understand that the methylation levels are different in AB and CB, but it would be nice if proliferation and differentiation levels could be measured after methylation or demethylation in cattle at the same time.

Response 2: Thank you very much for your suggestion. Another researcher in our group is conducting research related to methylation and demethylation. The focus of our research is to explore the effects of methylation from the tissue level.

Reviewer 3 Report

The authors present a manuscript discussing methylation of the FOXO1 promoter in myoblasts. While this topic will be of interest to researchers in muscle development, I have a few issues with how the results are presented.

* The authors use bisulphite PCR to identify methylation sites in the promoter. The results of this are shown in Figure 1. However, this figure is not clear or well described:

In panel A, the positions of the two CpG islands span ~1000–1700bp, whereas the label states that they are positioned at -366–1077. Please correct the axis to show the correct values.

Also in panel A, the positions indicated for the PCR primers do not overlap either of the CpG islands. Is this correct? There are also no MSP primers shown: either add them to the figure if they're missing, or remove them from the legend.

The descriptions of panels B and C seem to be swapped. Assuming that label C goes with panel B, the figure does not currently indicate which row corresponds to which age group. Also, the information in this panel would be useful in a supplementary table, with positions and methylation rates.

Section 3.2 only has one sentence, and doesn't explain how the results illustrated in Figure 2 lead to the conclusion that >90% of cells isolated were myoblasts.

Section 3.3 is also lacking detail. The abbreviation shRNA is not defined (I assume they are short hairpin RNAs, as used for RNAi, but this is not stated anywhere in the manuscript that I can find), and shNC is not defined anywhere: I guess that it's a negative control. Without any of this information, it is difficult to interpret these results.

Section 3.4: what do you mean by "value-addition"? Also, you need to define somewhere what the abbreviation CCK8 means, and what that assay is used for.

Author Response

Point 1. The authors use bisulphite PCR to identify methylation sites in the promoter. The results of this are shown in Figure 1. However, this figure is not clear or well described:

Response 1: Thank you for your suggestion. We revised Figure 1 to place both parts A and B in the supplementary material in order to improve the clarity of Figure 1, and modified the description in the results section of 3.1.

Point 2. In panel A, the positions of the two CpG islands span ~1000–1700bp, whereas the label states that they are positioned at -366–1077. Please correct the axis to show the correct values.

Response 2: Thank you for your suggestion. We have labeled Panel A and placed it in the supplementary material in order to improve the clarity of Figure 1. Since Panel A is generated directly by the online software we have no way to modify the corresponding information.

Point 3. Also in panel A, the positions indicated for the PCR primers do not overlap either of the CpG islands. Is this correct? There are also no MSP primers shown: either add them to the figure if they're missing, or remove them from the legend.

Response 3: Thank you for your suggestion. Due to the long bp of the CpG island, the online prediction software did not give us the ideal primers for BSP, so we redesigned the corresponding primer sequences using Primer Premier 5.0.

Point 4. The descriptions of panels B and C seem to be swapped. Assuming that label C goes with panel B, the figure does not currently indicate which row corresponds to which age group. Also, the information in this panel would be useful in a supplementary table, with positions and methylation rates.

Response 4: Thank you for your suggestion. We have revised the error that the descriptions of panels B and C were switched. We re-labeled Panel B in the original manuscript (which makes it clearer which row corresponds to which age group and the position of each CpG point corresponds to the mean methylation rate) and placed it in the supplementary material in order to improve the clarity of Figure 1.

Point 5. Section 3.2 only has one sentence, and doesn't explain how the results illustrated in Figure 2 lead to the conclusion that >90% of cells isolated were myoblasts.

Response 5: Thank you for your suggestion, we have revised it in the manuscript.

Point 6. Section 3.3 is also lacking detail. The abbreviation shRNA is not defined (I assume they are short hairpin RNAs, as used for RNAi, but this is not stated anywhere in the manuscript that I can find), and shNC is not defined anywhere: I guess that it's a negative control. Without any of this information, it is difficult to interpret these results.

Response 6: Thank you for your suggestion. We have added this explanation where it first appears in the manuscript (line 33 of shRNA: abstract and line 34 of shNC: abstract).

Point 7. Section 3.4: what do you mean by "value-addition"? Also, you need to define somewhere what the abbreviation CCK8 means, and what that assay is used for.

Response 7: Thank you for your suggestion. We have revised section 3.4 of the results in the original manuscript. We revised CCK8 to CCK-8 in the full text and added this explanation where it first appears (CCK-8: Simple Summary).